# PM2.5 Air Pollution Prediction through Deep Learning Using Multisource Meteorological, Wildfire, and Heat Data

**Pratyush Muthukumar** [1,*], **Kabir Nagrecha** [1], **Dawn Comer** [2], **Chisato Fukuda Calvert** [3], **Navid Amini** [1], **Jeanne Holm** [2] **and Mohammad Pourhomayoun** [1]

1. Department of Computer Science, California State University Los Angeles, Los Angeles, CA 90032, USA; knagrec2@calstatela.edu (K.N.); namini@calstatela.edu (N.A.); mpourho@calstatela.edu (M.P.)
2. City of Los Angeles, Los Angeles, CA 90012, USA; dawn.comer@lacity.org (D.C.); jeanne.holm@lacity.org (J.H.)
3. OpenAQ, Washington, DC 20009, USA; chisato@openaq.org
* Correspondence: pratyush.muthukumar@gmail.com

**Abstract:** Air pollution is a lethal global threat. To mitigate the effects of air pollution, we must first understand it, find its patterns and correlations, and predict it in advance. Air pollution is highly dependent on spatial and temporal correlations of prior meteorological, wildfire, and pollution structures. We use the advanced deep predictive Convolutional LSTM (ConvLSTM) model paired with the cutting-edge Graph Convolutional Network (GCN) architecture to predict spatiotemporal hourly PM2.5 across the Los Angeles area over time. Our deep-learning model does not use atmospheric physics or chemical mechanism data, but rather multisource imagery and sensor data. We use high-resolution remote-sensing satellite imagery from the Moderate Resolution Imaging Spectroradiometer (MODIS) instrument onboard the NASA Terra+Aqua satellites and remote-sensing data from the Tropospheric Monitoring Instrument (TROPOMI), a multispectral imaging spectrometer onboard the Sentinel-5P satellite. We use the highly correlated Fire Radiative Power data product from the MODIS instrument which provides valuable information about the radiant heat output and effects of wildfires on atmospheric air pollutants. The input data we use in our deep-learning model is representative of the major sources of ground-level PM2.5 and thus we can predict hourly PM2.5 at unparalleled accuracies. Our RMSE and NRMSE scores over various site locations and predictive time frames show significant improvement over existing research in predicting PM2.5 using spatiotemporal deep predictive algorithms.

**Keywords:** air pollution prediction; spatiotemporal forecasting; deep convolutional LSTM; remote-sensing satellite imagery; wildfire heat data; meteorological data

## 1. Introduction

Air pollution is a destructive global crisis. It has a death toll of 7 million people per year, of which 600,000 are children [1]. The harmful effects of short-term and long-term exposure to air pollution decrease the global life expectancy by 1–2 years on average [2]. Over one in four deaths of children under the age of five can be directly traced back to the deadly effects of air pollution [3]. Air pollution is linked to various adverse health effects such as asthma, emphysema, cardiovascular illness, and respiratory illness. Within Los Angeles, there are over 27 million tons of atmospheric nitrogen dioxide, which is 1.5 times the amount of the next leading U.S. city [4]. It is evident that finding an effective and reliable solution to reducing ambient air pollution will drastically improve global health and wellbeing.

To mitigate the deadly effects of air pollution, we must first be able to understand it, discover its causes and patterns, and predict it in advance. This paper describes our approach using deep predictive models and advanced machine-learning algorithms to learn patterns of spatiotemporal air pollution in various locations and predict for the future.

When developing these cutting-edge high-performance deep-learning models, we focus on learning correlations of both the spatial and temporal patterns in the data. Air pollution prediction is inherently a spatiotemporal task: air pollutants travel in the air and thus affect the surrounding areas (spatial correlation); air pollution concentrations in the future depend on prior concentrations (temporal correlation).

Air pollution prediction has been a topic of interest for decades, with the most recent approaches focusing on using the predictive capabilities of deep neural networks; see the survey paper Bellinger et al. [5] and the references therein. Current deep-learning research in this field seeks to learn and predict either the spatial patterns or temporal patterns of ambient air pollution, but we seldom see models capable of learning and predicting both [6–9]. Of the limited works performing spatiotemporal prediction, the current state-of-the-art deep-learning models have lower spatial and temporal resolution than the model we propose in this paper. Zhang et al. [10] proposes a spatiotemporal prediction model for predicting daily PM2.5 with a spatial resolution of $3.3 \times 3.3$ km per unit prediction with a mean RMSE of 14.94 $\mu g/m^3$ for 28th-day prediction. We propose a model capable of learning the spatial and temporal correlations of air pollution measured through hourly multisource big data at a spatial resolution of $1 \times 1$ km.

For effective and accurate prediction with deep learning, the key defining tenet of a successful model is the quality and heterogeneity of the input data. We use meteorological ground-based sensor data, air pollutant remote-sensing satellite imagery, wildfire/heat data, and ground-based pollutant sensor data from high-quality validated monitoring sites in our model to deliver state-of-the-art accuracy in our predictions.

The novelty of our approach is evident in the multisource big data sources and complex deep-learning architectures employed to learn and predict spatiotemporal correlations in air pollution data. Our novel approach of sequentially learning meteorological feature correlations through a complex Graph Convolutional Network (GCN) architecture and inputting the learned representations into a ConvLSTM architecture alongside wildfire data, remote-sensing satellite imagery, and ground-based air pollutant sensor data is the first of its kind in tackling the spatiotemporal air pollution prediction problem in this field.

We make the following contributions: (1) we propose a novel deep predictive GCN model architecture capable of effectively interpolating, learning, and predicting spatiotemporal patterns in meteorological data; (2) we collect and use a comprehensive multisource meteorological, wildfire, atmospheric air pollutant, and ground-based PM2.5 dataset for prediction; (3) we develop a deep-learning pipeline using the cutting-edge GCN and ConvLSTM models to predict spatiotemporal PM2.5 in Los Angeles county with state-of-the-art accuracies.

The remainder of the paper is structured as follows. Section 2 describes our methodology. Sections 2.1 and 2.3 describes our model architecture and implementation. Section 3 describes our model's experimental results. Section 4 concludes our findings and Section 5 discusses future work.

## 2. Methodology

In the following section, we describe our methodology for constructing our two-stage model for predicting spatiotemporal PM2.5 pollutants over Los Angeles county. Our motivations for using multisource meteorological data, wildfire data, remote-sensing satellite imagery, and ground-based air pollution sensor data are that these data sources are interrelated, and including data from all sources provides the necessary depth of clarity to accurately predict spatiotemporal air pollution.

We employ a sequential two-stage model to extract the learned representations from the concentrated input data that we use to predict spatiotemporal particulate matter 2.5 (PM2.5) in various areas of Los Angeles county over time. PM2.5 is denoted as particulate matter pollutants with a diameter of less than 2.5 micrometers. PM2.5 is perhaps the deadliest air pollutant on Earth: according to McGill University [11], heightened levels of PM2.5 has the strongest correlation to early deaths of humans than any other air pollutant. The first stage of our model uses the cutting-edge Graph Convolutional Network

(GCN) model to learn and predict patterns between meteorological and ground-based PM2.5 sensor data. The second stage is sequentially fed the outputs of the first stage in addition to wildfire/heat data and remote-sensing satellite imagery of various atmospheric air pollutants to predict hourly PM2.5 over various locations of Los Angeles county in the future.

Predicting air pollution in the geographical location of Los Angeles also poses unique challenges. The Los Angeles basin is almost completely enclosed by mountains to the north and east. The vertical temperature structure (inversion) tends to prevent vertical mixing of the air through more than a shallow layer (1000 to 2000 feet deep). Moreover, the southern location of the LA basin permits a fairly regular daily reversal of wind direction—offshore at night and onshore during the day. Finally, the metropolitan city of Los Angeles and concentrated population leads to atmospheric pollution accumulating and remaining within this circulation pattern [12].

*2.1. Model Architecture*

The Graph Convolutional Network (GCN) is a complex deep-learning architecture applied upon graphs [13]. Graphs are a valuable and effective method of modeling air pollution and weather forecasting, since the bulk of open-access air pollution and meteorological data are in the form of stationary ground-based sensors. Thus, it is intuitive to draw a parallel between these ground-based sensors and nodes in a weighted directed graph. A weighted directed graph provides the additional functionality to preserve the spatial and distance-based correlations among sensors. The goal of the Graph Convolutional Network is to learn the feature embeddings and patterns of nodes and edges in a graph. The GCN learns the features of an input graph $G(V, E)$ typically expressed with an adjacency matrix $A$ as well as a feature vector $x_i$ for every node $i$ in the graph expressed in a matrix of size $V \times D$ where $V$ is the number of vertices in the graph and $D$ is the number of input features for each vertex. The output of the GCN is an $V \times F$ matrix where $F$ is the number of output features for each vertex. We can then construct a deep neural network with an initial layer embedding of $h_v^0 = x_i$ to perform convolution neighborhoods of nodes, similar to a Convolutional Neural Network (CNN). Then, the $k$-th layer of the neural network's embedding on vertices $h_v^k$ is

$$h_v^k = \sigma\left( W_k \sum_{u \in N(v) \cup v} \frac{h_v^{k-1}}{\sqrt{|N(u)||N(v)|}} \right), \forall k > 0,$$

where $\sigma$ is some non-linear activation function, $h_v^{k-1}$ is the previous layer embedding of $v$, $W_k$ is a transformation matrix for self and neighbor embeddings, and $\sum_{u \in N(v)} \frac{h_u^{k-1}}{|N(v)|}$ is the average of a neighbor's previous layer embeddings. The neural network can be trained efficiently through sparse batch operations on a layer wise propagation rule

$$H^{(k+1)} = \sigma(D^{-\frac{1}{2}} \tilde{A} D^{-\frac{1}{2}} H^{(k)} W_k),$$

where $I$ is the identity matrix, $\tilde{A} = A + I$, and $D$ is the diagonal node degree matrix defined as $D_{ii} = \sum_j A_{i,j}$ [14]. In this way, the GCN can train a neural network to output a graph with feature vectors for each node in the graph. In our implementation, we extend the GCN model's capabilities further by providing a feature matrix constructed of feature vectors for each edge in the graph such that the GCN outputs a graph with an output feature matrix for all nodes and edges in the graph.

The second stage of our model uses the highly effective Convolutional Long Short-Term Memory (ConvLSTM) deep-learning architecture which learns and predicts for data considering both spatial and temporal correlations. The ConvLSTM model is a variant of the traditional Long Short-Term Memory (LSTM) model, a time-series Recurrent Neural Network (RNN).

Traditional LSTM models rely on a single-dimensional input vector parameterized by time. The structure of the LSTM model relies on a recurrent sequential architecture of gates and cells which retain and propagate certain information from previous cells and data. For a traditional FC-LSTM (Fully Connected Long Short-Term Memory), the time-parameterized input gates $i_t$, forget gates $f_t$, cell states $c_t$, output gates $o_t$, and hidden gates $h_t$ are defined as

$$i_t = \sigma(W_i x_t + W_i h_{t-1} + W_i \circ c_{t-1} + b_i)$$
$$f_t = \sigma(W_f x_t + W_f h_{t-1} + W_f \circ c_{t-1} + b_f)$$
$$c_t = f_t \circ c_{t-1} + i_t \circ \tanh(W_x x_t + W_h h_{t-1} + b_c)$$
$$o_t = \sigma(W_x x_t + W_h x_{h-1} + W_c \circ c_t + b_o)$$
$$h_t = o_t \circ \tanh(c_t),$$

where $W$ denotes the weight matrix and $\circ$ denotes the Hadamard matrix multiplication product [15]. In a traditional FC-LSTM, both the inputs and outputs are single-dimensional time-series vectors. As a result, LSTM models do not allow for or use spatial correlations in data. More generally, the traditional FC-LSTM architecture does not allow image or video-like inputs.

The ConvLSTM model improves upon the FC-LSTM by applying convolution within the cells and gates of the LSTM to allow for multidimensional video-like inputs and outputs. This can be achieved by replacing the Hadamard products used to define the key equations for the FC-LSTM with the convolution operation. Intuitively, this replacement effectively serves as an intermediary processing layer between the video-like input and the traditional LSTM model by transforming the video-like input frames to single-dimensional vectors through convolution at each cell of the FC-LSTM. The key equations for the ConvLSTM are

$$i_t = \sigma(W_i x_t + W_i h_{t-1} + W_i * c_{t-1} + b_i)$$
$$f_t = \sigma(W_f x_t + W_f h_{t-1} + W_f * c_{t-1} + b_f)$$
$$c_t = f_t * c_{t-1} + i_t * \tanh(W_x x_t + W_h h_{t-1} + b_c)$$
$$o_t = \sigma(W_x x_t + W_h x_{h-1} + W_c * c_t + b_o)$$
$$h_t = o_t * \tanh(c_t),$$

where $*$ denotes the convolution operation [16].

Please note that there are two methods to induce convolution in a traditional LSTM model. One such method is denoted as the ConvLSTM model and uses the convolution operation within the cells and gates of the LSTM, thus directly allowing the inputs and outputs of the ConvLSTM to be time-series multidimensional data. Another method of inducing convolution is to perform convolution prior to and separately from the LSTM model. By modularizing the convolution operation and first training a Convolutional Neural Network (CNN) to transform video-like inputs to single-dimensional time-parameterized output vectors and then using the CNN's output in a traditional FC-LSTM, we can achieve a similar level of learning and prediction based on spatial and temporal correlations. This approach is succinctly presented as a standalone deep-learning model denoted the Convolutional Neural Network—Long Short-Term Memory (CNN-LSTM), which, as the name suggests, uses a CNN and LSTM run in series to learn and predict video-like inputs. We perform spatiotemporal air pollution prediction using the ConvLSTM model; however, there is prior research on alternatively using the CNN-LSTM model to predict spatiotemporal air pollution [17–20].

More specifically, we find that including meteorological features is essential to an accurate prediction of ambient air pollution. Air pollutants are closely correlated with meteorological data. A recent study found that of the 896 government-monitored air pollution sensors in China, 675 ground-based sensors reported an increase in carbon

monoxide ($CO$), sulfur dioxide ($SO_2$), nitrogen dioxide ($NO_2$), and PM2.5 when there was a greater than 10% increase in wind speed at the same location [21].

In the geographical setting of Los Angeles county, it comes as no surprise that wildfire/heat data can provide useful insights into the structures and patterns of atmospheric air pollutants. We find that including remote-sensing satellite imagery and ground-based grid sensor data information on the wildfire, smoke, and heat patterns in Los Angeles county greatly improved the accuracy of our predictive model for forecasting PM2.5. In fact, Burke et al. [22] found that wildfire smoke now accounts for up to half of all fine-particle pollution including PM2.5 in the Western U.S and up to 25% of fine-particle pollution nationwide. Thus, a major focus of our model is fixated on effectively using the wildfire/smoke information for the reliable and accurate prediction of PM2.5 in Los Angeles county.

We find that including a mixture of both remote-sensing satellite imagery of air pollution and ground-based sensor air pollution data is necessary for a robust and multifaceted approach to spatiotemporal air pollution prediction. Remote-sensing satellite imagery provides information on atmospheric air pollution and its general structures, while ground-based sensors provide finer-grained information on air pollution at sea level or within cities. Since the level of air pollution may vary greatly with respect to altitude, we use both remote-sensing satellite imagery and ground-based sensor data as input to our model to fully understand and predict air pollution.

Finally, we find that including remote-sensing satellite imagery and ground-based sensor data from other air pollutants prove to be beneficial when predicting for a particular pollutant—in our case PM2.5. For example, although the goal of our model is to predict PM2.5, we include imagery and sensor data of nitrogen dioxide ($NO_2$), carbon monoxide (CO), ozone ($O_3$), and methane ($CH_4$) atmospheric air pollutants because these pollutants are closely linked to PM2.5 [23]. According to Jiao and Frey [24], the main source of both PM2.5 and atmospheric carbon monoxide is vehicle exhausts and emissions. Thus, it serves only benefit to include additional air pollutant data when predicting for spatiotemporal PM2.5.

The first stage of our model uses the Graph Convolutional Network architecture to learn patterns of meteorological data through a graph representation. We first construct a weighted directed graph representation with the meteorological data. The goal of the GCN architecture is to interpolate a denser meteorological graph with more nodes and connecting edges than the input graph. The primary issue with high-quality validated ground-based meteorological sensor data is that for a geographic area as fine-grained as Los Angeles county, the number of site locations and meteorological features are sparse. Basic interpolation techniques such as distance-weighted interpolation and nearest neighbor interpolation fail to accurately map the spatiotemporal correlations of meteorological data. The task of interpolation is inherently an effective task to obtain high-level learned feature embeddings. By applying a GCN architecture for spatial interpolation, we can train a deep-learning model to predict meteorological trends in areas not provided by the input graph. We can later use these interpolated correlations as inputs to construct a video-like sequence of spatially continuous predicted meteorological features over time in our geographical area. For our model, we adapt previous work on spatiotemporal kriging with Graph Convolutional Networks to interpolate our nodes and edges of the meteorological graph [25]. We train the GCN for this interpolation task by systematically hiding a small percentage of node and edges and their corresponding attribute vectors. The GCN model learns to predict for the hidden meteorological node and edge feature values using the ground truth data from a neighborhood of nodes and edges surrounding the missing information. By iteratively training the interpolation process using the loss function between the ground truth hidden attribute values and the predicted values, the GCN can interpolate a sparse meteorological graph into a dense graph containing various meteorological features. The GCN will create a dense meteorological graph for each sample parameterized by time. In the case of our model, the GCN interpolates the sparse meteorological graph into a dense graph for every hour of the hourly meteorological dataset.

We provide a visualization of this interpolation training process in Figure 1. We visualize two frames of the interpolation training process on the meteorological graph structure for a single stationary attribute of AQI.

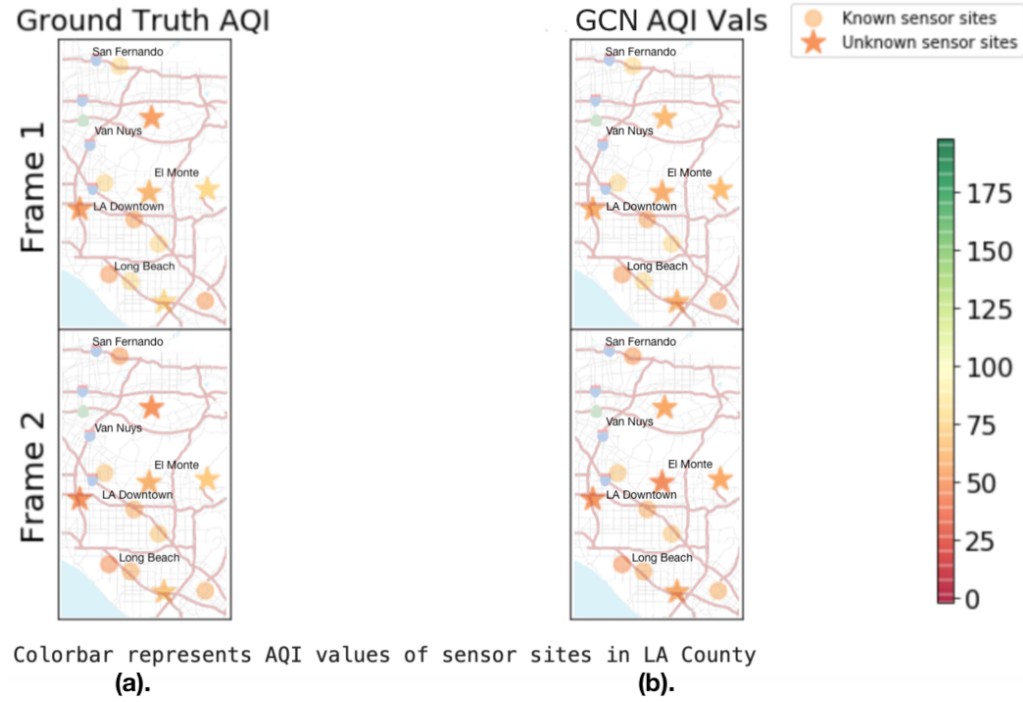

**Figure 1.** AQI Node Attribute Training Prediction Visualization. (**a**) shows the Ground Truth AQI node attribute values for two frames an hour apart, (**b**) shows the GCN Predicted AQI node attribute values for two frames an hour apart.

An intermediate step in our model converts the GCN-interpolated dense meteorological graph into an image-based format and concatenates many time-series samples into a video-like input to the ConvLSTM model. We apply an unsupervised learning graph representation learning approach to create a matrix of high-level weights corresponding to the representations of nodes and edges in the meteorological graph. This set of weights is bounded by the geographic area we have defined, and as a result, the high-level embedding weight array is calculated for each timestep of the meteorological dataset. By converting the dense meteorological graphs into spatiotemporal embeddings in a video-like input, we can pass the learned meteorological information as input to the second stage of our model. A visualization of the first stage and the intermediate step to convert raw meteorological data into high-level embeddings from dense interpolated graphs is described in Figure 2.

The second stage of our model uses the ConvLSTM architecture to predict spatiotemporal PM2.5. The inputs to the ConvLSTM model are all video-like in format: all input data are formatted as frames of images or arrays parameterized over time. The inputs to the ConvLSTM model are the learned meteorological information outputs from the first stage of the model, the remote-sensing satellite imagery of air pollutants, the wildfire heat data, and the ground-based sensor data of air pollutants. The output of the ConvLSTM model is a set of predicted ground-based PM2.5 sensor values around Los Angeles county for multiple days in the future. Figure 3 displays a visualization of the ConvLSTM architecture which makes up the second module of our model.

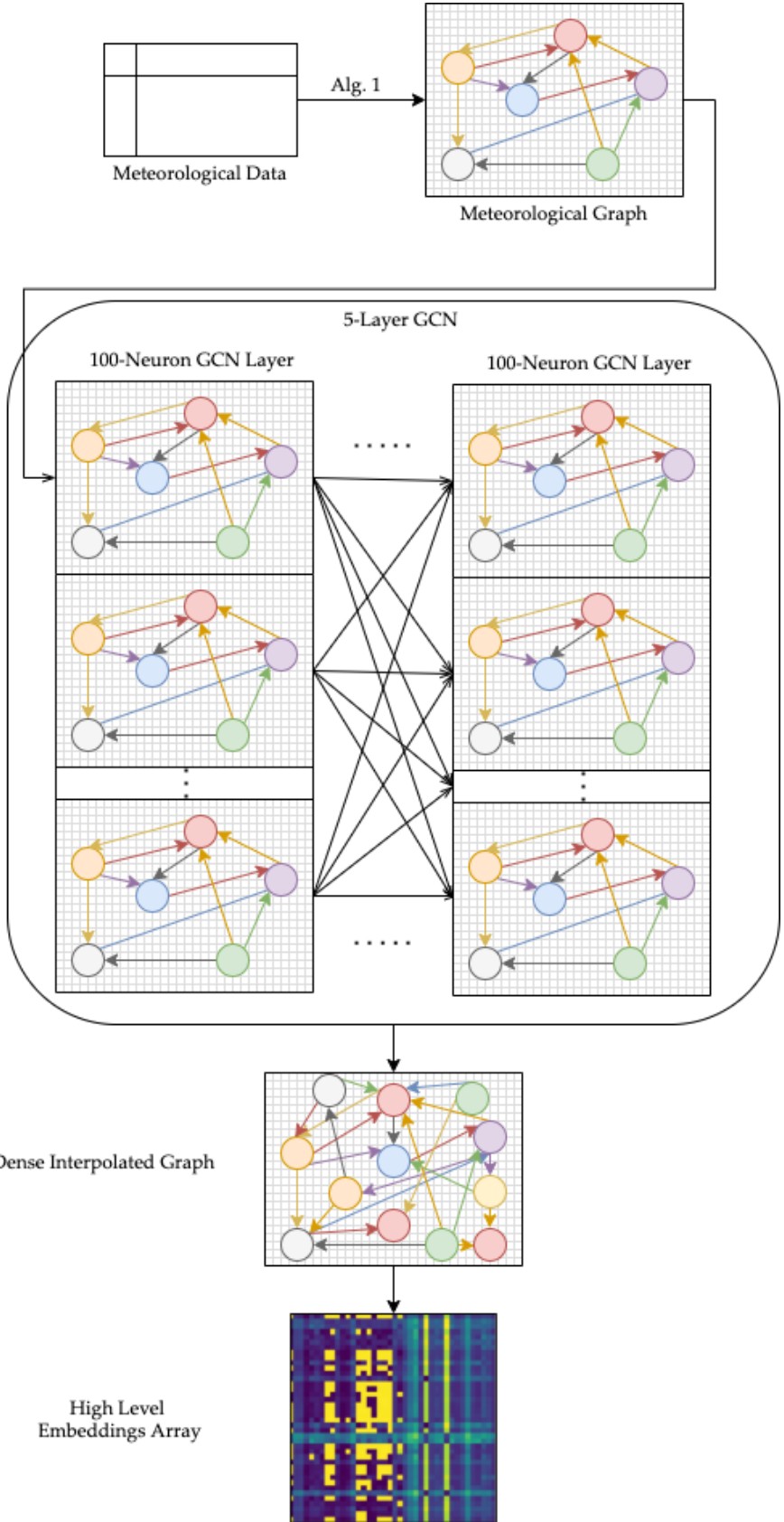

**Figure 2.** Visualization of GCN module of our model architecture applied to meteorological data.

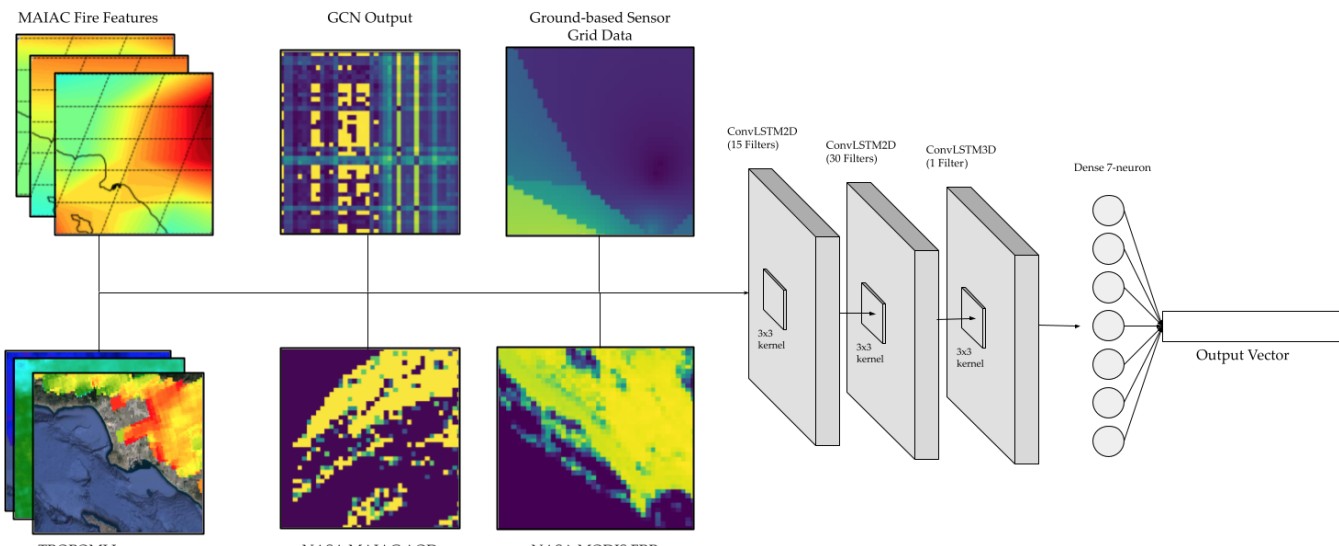

**Figure 3.** Visualization of the ConvLSTM module of our model architecture.

*2.2. Dataset*

Our geographical bounds for prediction is a square region of roughly 2500 mi² of northwest Los Angeles county. More specifically, we select the square region with corner coordinates ranging from 33.5° N to 34.5° N and 117.5° W to 118.75° W. We format all input data to fit these geographical bounds. For remote-sensing satellite imagery in our dataset, we crop the satellite images to fit the geographic boundaries we defined. For the ground-based sensors, we use the data from all sensors within the latitude and longitude range of our geographic boundary.

Our temporal bound for prediction is three years of data from 1 January 2018 to 31 December 2020. Each sample of our dataset has an hourly temporal frequency. This hourly frequency is standard across all input data and prediction results. For each of our data sources, we collect 26,304 samples corresponding to 24 hourly samples for the 1096 days of data from 1 January 2018 to 31 December 2020.

Our meteorological data are collected from the Iowa State University Environmental Mesonet database [26]. The Environmental Mesonet database collects and records hourly Meteorological Aerodrome (METAR) Reports from Automated Surface Observing Systems (ASOS) located near various airports and municipal airstrips within the continental United States. The ASOS data are primarily used by airlines and air traffic controllers to monitor meteorological features near and around airport runways. The METAR data provides comprehensive hourly reports of 17 ground-level meteorological features including wind speed, wind direction, relative humidity, dew point, precipitation, Air Quality Index (AQI), air pressure, and air temperature. The complete list of meteorological features collected from each site is presented in the Appendix A (Table A2). Within our geographic boundaries, there are 24 ASOS sensors providing comprehensive, validated, and quality checked METAR reports. Figure 4 describes the geographical area of interest and site locations for the raw meteorological features we collected.

To use these meteorological features within the model, we must transform the array format of the meteorological features into hourly meteorological graphs for the GCN model. First, since each of the meteorological features are recorded in terms of their respective units, we normalize the various units of these meteorological features. To normalize the units, we calculate each data point's percentile value with respect to the previous day's maximum value. This percentile value is calculated for each hourly sample and is the ratio between the current sample's raw value and the metric's maximum value across all 24 samples for the previous day. In this way, we retain the important meteorological

information relative to each metric without relying on the domain-specific units of each meteorological feature.

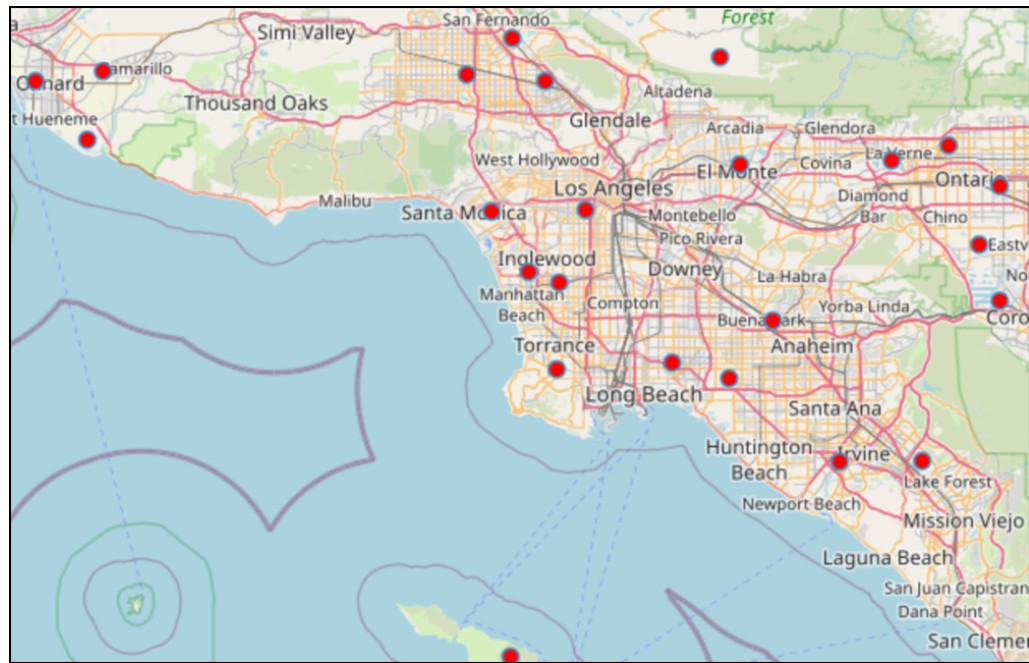

**Figure 4.** METAR ASOS data from the Mesonet database: 24 sensor locations in Los Angeles county, where each sensor records 17 hourly meteorological attributes.

Our ground-based sensor PM2.5 dataset is collected from the Southern California Air Resources Board AQMIS2 portal [27]. For the geographic range we have defined, there are seven quality-assured, validated PM2.5 monitoring sites collecting hourly data in the following locations: Lancaster, Santa Clarita, Reseda, Glendora, Los Angeles—North Main St, Long Beach, and Long Beach—Rt 710. These seven PM2.5 sensors are the only government-maintained PM2.5 sensors within the geographical bounds; however, there are various low-cost privately maintained sensors we did not use in our predictive model. To effectively validate the performance of our model, we select only highly regulated and closely maintained sensors to ensure that the error uncertainty for the raw sensor measurements is as low as possible. We use historical PM2.5 data at these locations while training the model and validate the accuracy of our model by measuring the error between our predicted PM2.5 values for future timesteps at these locations against the ground truth PM2.5 values.

Our remote-sensing data of various air pollutants is collected from the NASA Multi-Angle Implementation of Atmospheric Correction (MAIAC) algorithm and the ESA TRO-POspheric Monitoring Instrument (TROPOMI) data sources [28,29]. The MAIAC algorithm is a preprocessing algorithm performed on imagery collected by the NASA Moderate Resolution Imaging Spectroradiometer (MODIS) instrument onboard the NASA Terra and Aqua satellites [30]. The Terra and Aqua provide imagery over 36 spectral bands using the MODIS imaging instrument. The MAIAC algorithm is a complex data-preprocessing algorithm that converts raw MODIS imagery to data analytics ready samples by retrieving atmospheric aerosol and air pollutant data from MODIS images, normalizing pixel values, interpolating daily data for hourly use, and removing cloud cover masks.

In our model, one of the remote-sensing satellite imagery collections we use is the MAIAC MODIS/Terra+Aqua Daily Aerosol Optical Depth (AOD) dataset. AOD is a measure of the direct amount of sunlight blocked by atmospheric aerosols and air pollutants. This measure is perhaps the most comprehensive measure of ambient air pollution, and years of research has shown a strong correlation between AOD readings and PM2.5 concentrations in both atmospheric and ground-level settings [31,32]. The MAIAC MODIS AOD dataset we use in

our predictive model records the blue-band Aerosol Optical Depth at a central wavelength of 0.47 μm. The raw MAIAC MODIS AOD dataset provides a spatial resolution of 1 km/pixel for an area of 1200 km by 1200 km. For our implementation, we crop the imagery to fit our defined geographic bounds within Los Angeles county.

Figure 5 describes a sample of NASA MODIS AOD imagery after preprocessing from the MAIAC algorithm. We also apply an additional preprocessing step to downsample the MAIAC output to a grid of 40 by 40 pixels within our geographic bounds. This down sampling is performed to normalize the sample sizes across all input sources. Please note that the figure provides a visualization of the raw grid-like data of the MAIAC AOD imagery, and thus the color values of the visualization correspond to AOD values, not raw RGB values. As such, this visualization's color values should be interpreted as a color map, not as a visual indicator of true AOD values. The brighter-colored pixels in the visualization correspond to higher AOD values.

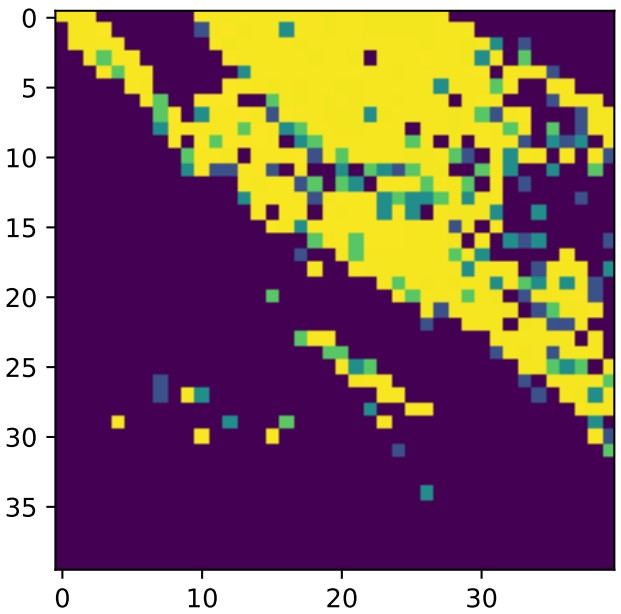

**Figure 5.** Example Downsampled MAIAC Satellite AOD Imagery (1 January 2019 within Los Angeles).

We also collect remote-sensing data from the TROPOMI instrument onboard the ESA Sentinel-5P satellite. The Sentinel-5P satellite launched on 13 October 2017, orbiting at a height of 512 miles above sea level, with an orbital swath of 2600 km, and a mission length of seven years (2017–2024). The Sentinel-5P TROPOspheric Monitoring Instrument (TROPOMI) is a spectrometer capable of sensing ultraviolet (UV), visible (VIS), near (NIR) and short-wavelength infrared (SWIR) light. TROPOMI provides high-resolution global hourly data of atmospheric ozone, methane, formaldehyde, aerosol, carbon monoxide, nitrogen dioxide, and sulfur dioxide. For our model, we use remote-sensing data of methane ($CH_4$), nitrogen dioxide ($NO_2$), and carbon monoxide ($CO$). We chose these air pollutants based on its correlation to PM2.5 and the spatial resolution of TROPOMI data. For these features, we apply additional downsampling to the TROPOMI data to generate hourly 40-by-40-pixel grids of data for each air pollutant. Figure 6 describes examples of the downsampled methane, nitrogen dioxide, and carbon monoxide data used in our model.

We use ground-based and atmospheric wildfire and heat data to predict spatiotemporal PM2.5 in Los Angeles county. We collect wildfire and heat data from two sources: NASA MODIS data and NASA MERRA-2 data. We collect Fire Radiative Power (FRP) imagery from the NASA MODIS/Terra Land Surface Temperature and Emissivity collection [33]. Fire Radiative Power (FRP) is a measure of the radiant heat output from a fire. The main contributors to increased levels of FRP include smoke from wildfires and emissions from

the burning of carbon-based fuel, such as carbon monoxide ($CO$) and carbon dioxide ($CO_2$) emissions. Thus, there is a strong positive correlation between wildfires and FRP values as well as a weaker positive correlation between carbon emissions ($CO_2$, $CO$) and FRP values. FRP is measured in megawatts (MW) and can be collected using an imaging instrument onboard a remote-sensing satellite aircraft. The wavelength of light needed to image FRP is in the range of 2070 μm to 3200 μm.

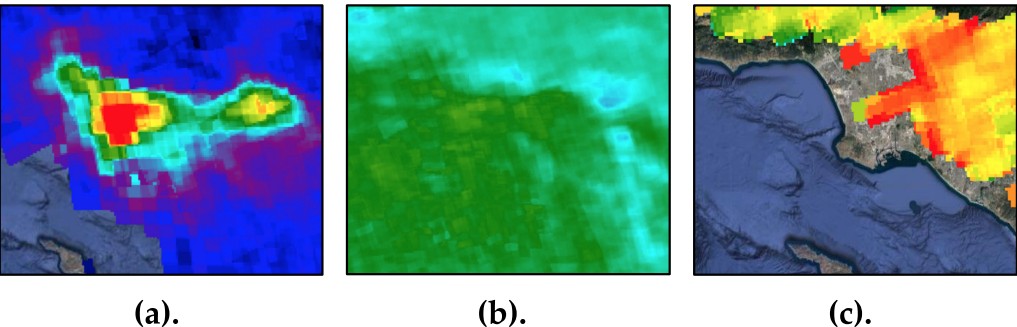

**Figure 6.** Downsampled TROPOMI remote-sensing data of various air pollutants (1 January 2019 within Los Angeles), (**a**) nitrogen dioxide, (**b**) carbon monoxide, (**c**) methane.

In our model, we find that including FRP imagery drastically improves our model's performance during the wildfire seasons such as months from June through November when predicting for spatiotemporal PM2.5 in Los Angeles. We also see improvements in our model's performance in non-winter months after including FRP, since during these months, the imagery provides information on carbon-based fuel emissions, which is highly correlated with the movement and structure of PM2.5 [34]. Figure 7 describes a 40-by-40-pixel downsampled visualization of FRP values over various times of the year. Again, note that the figure provides visualizations of the raw grid-like data of the MODIS FRP imagery, and thus the color values of the visualization correspond to FRP values, not raw RGB values.

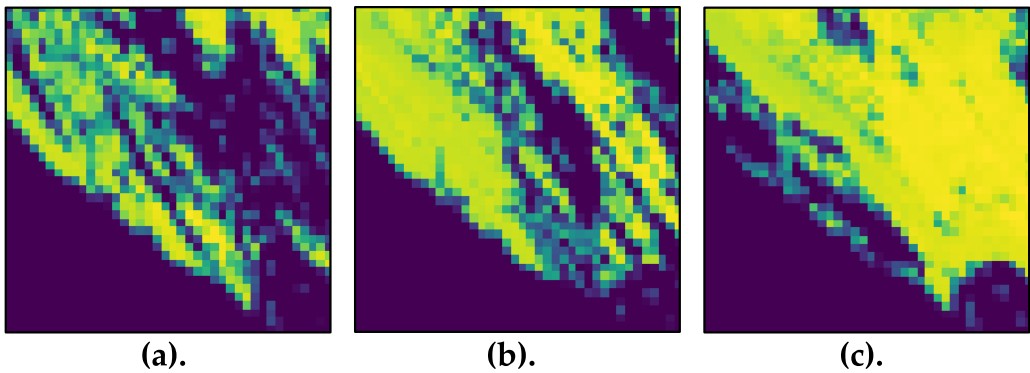

**Figure 7.** Downsampled FRP visualization samples at various times in Los Angeles, (**a**) March 2018, (**b**) June 2018, (**c**) August 2018.

We also use wildfire and heat data from the NASA MERRA-2 data source. The Modern-Era Retrospective analysis for Research and Applications, version 2 (MERRA-2) is a global atmospheric reanalysis produced by the NASA Global Modeling and Assimilation Office (GMAO). It spans the satellite observing era from 1980 to the present. The goals of MERRA-2 are to provide a regularly gridded, homogeneous record of the global atmosphere, and to incorporate additional aspects of the climate system including trace gas constituents (stratospheric ozone), and improved land surface representation, and cryospheric processes [35]. All the MERRA-2 features we use in our predictive model are in the format of multidimensional arrays of grid-based raw values throughout Los Angeles over time.

We use MERRA-2 imagery of three wildfire/heat features: Planetary Boundary Layer (PBL) height, surface air temperature, and surface exchange coefficient for heat. Planetary Boundary Layer Height is a measure of the distance from ground level of the lowest part of the atmosphere. The lowest part of the atmosphere, or the peplosphere, is directly influenced by the changing surface temperature of Earth, various aerosols in the atmosphere, and is especially influenced by smoke or ash from a fire. PBL height is also influenced by precipitation and changes in surface pressure. Over deserts or areas of dry, warm climates that may be caused by fires burning in the area, the PBL may extend up to 4000 to 5000 m above sea level. Over cooler, more humid temperatures with little aerosols, dust, or smoke in the atmosphere, the PBL may be less than 1000 m above sea level. Thus, intuitively, a low PBL height means that there are no fires burning in the area and the atmosphere is relatively clear of aerosols. Figure 8 provides a visualization of the MERRA-2 imagery for PBL height globally and over Los Angeles county.

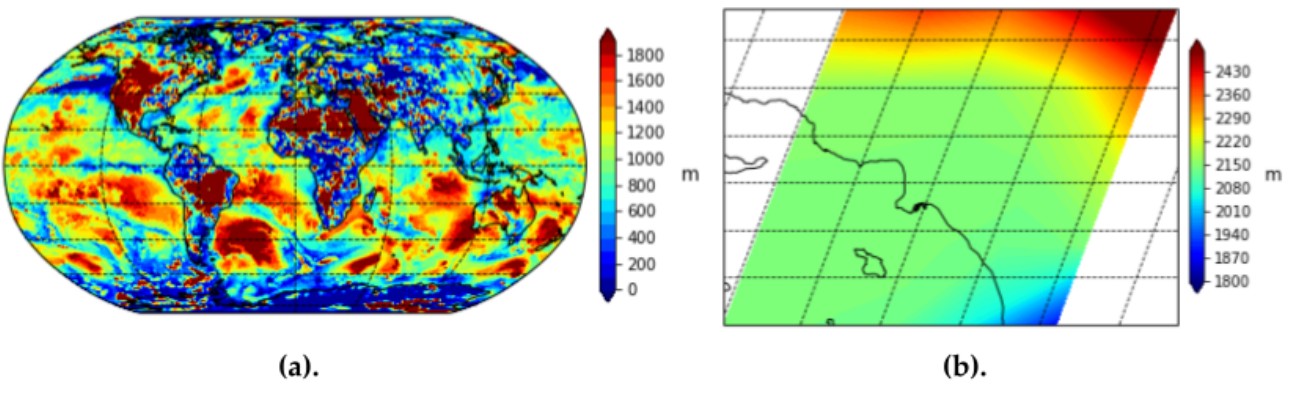

**Figure 8.** MERRA-2 PBL Height (in meters) visualization over various geographic scales, (**a**) Global, (**b**) Los Angeles.

We also use the surface air temperature feature collection from MERRA-2 to provide general information about heat. We find that wildfires, smoke plumes, and industrial exhausts will all influence surface air temperature. Figure 9 provides a visualization of the MERRA-2 imagery for surface air temperature globally and over Los Angeles county.

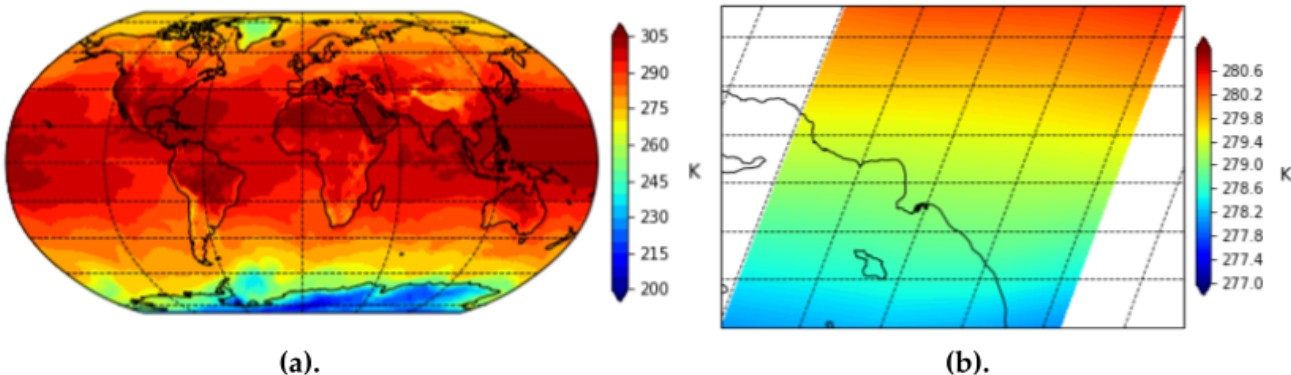

**Figure 9.** MERRA-2 Surface Air Temperature (in Kelvin) visualization over various geographic scales, (**a**) Global, (**b**) Los Angeles.

Finally, we use the MERRA-2 surface exchange coefficient for heat features. Surface exchange coefficient for heat provides insights into the effects of wildfires, heat, and smoke plumes over non-terrain regions as well, such as oceans and rivers. The surface exchange coefficient for heat feature provides especially useful information since our geographical area of interest is Los Angeles. For example, we find that including surface exchange

coefficient for heat helped the model understand atmospheric PM2.5 over the Pacific Ocean near the Port of Los Angeles and Port of Long Beach. Figure 10 provides a visualization of the MERRA-2 imagery for surface exchange coefficient for heat globally and over Los Angeles county. We provide the full summary of input data used, the datasets we collected them from, the instruments used to capture the data, and the data source types in the Appendix A (Table A1).

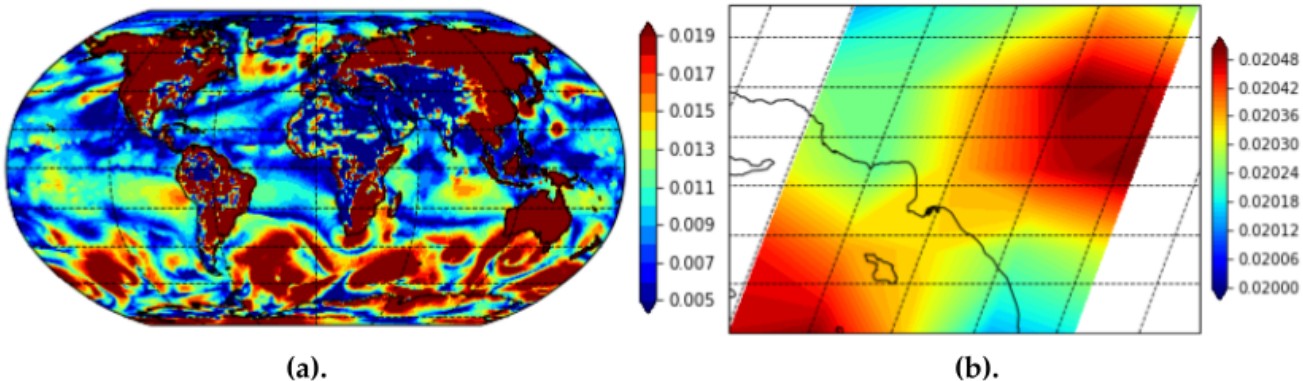

**Figure 10.** MERRA-2 Surface Exchange Coefficient for Heat visualization over various geographic scales, (**a**) Global, (**b**) Los Angeles.

*2.3. Implementation*

In our GCN architecture which uses meteorological data to create dense interpolated high-level embeddings of meteorological features, we must preprocess raw meteorological data into weighted directed graphs with node and edge attribute vectors. That is, for each timestep of the meteorological dataset, we create a weighted directed graph denoting the nodes of the graph as "stationary" meteorological features pertaining to a sensor location and the edges denoting "non-stationary" meteorological features. We define "stationary" features as scalar measurements of individual meteorological features at a sensor location. For example, the node attributes for our meteorological graph include relative humidity, AQI, temperature, air pressure, dew point, and heat index. We describe the full dichotomy of "stationary" and "non-stationary" attributes in the Appendix A (Table A2). Edge attributes consist of "non-stationary" meteorological features that rely on or connect multiple sensors. For example, the edge attributes consist of the physical distance in miles from meteorological sensor locations, the wind speed, and the wind direction. For each timestep, we can create a multidimensional weighted directed graph containing the spatial and distance-based information of all meteorological sensors and their recorded features. We then repeat this process to create these multidimensional weighted directed graphs for each hourly sample in the dataset. Algorithm 1 describes a step-by-step procedure of creating these weighted directed meteorological graphs for a single timestep.

To implement the ConvLSTM architecture, we use the Keras ConvLSTM layer [36]. This implementation requires the input data to be in the form of a five-dimensional tensor with dimensions (sample, frame, row, column, filter). For the remote-sensing satellite imagery in our dataset, we set the row, column, and filter dimensions as the 2D image along with the RGB color values as the filter. All remote-sensing satellite imagery data sets are downsampled to 40-by-40-pixel resolutions, which correspond to a 40 row by 40 column array for the 5D tensor input. For the ground-based sensor data, we create a 40-by-40-pixel grid and use the latitude and longitude coordinates of the monitoring sites to set the location of the sensor values within the array, similar to the process described in Algorithm 1.

---

**Algorithm 1** Meteorological Graph Construction

---

**Input:** Meteorological site features $f_i \in F$, where each $f_i$ contains site coordinates $x_i, y_i$ and a set of site-specific stationary $s_i \in S$ and non-stationary $n_i \in N$ feature values. Boundary latitude values $\text{lat}_{max}, \text{lat}_{min}$. Boundary longitude values $\text{long}_{max}, \text{long}_{min}$.

    Initialize $40 \times 40$ array grid A.
    Initialize weighted directed graph $G = (V, E)$
    **for** $f_i \in F$ **do**
        $\text{grid}_x, \text{grid}_y = \left\lfloor \frac{x_i \cdot 40}{\text{long}_{max} - \text{long}_{min}} \right\rfloor, \left\lfloor \frac{y_i \cdot 40}{\text{lat}_{max} - \text{lat}_{min}} \right\rfloor$
        $\text{A}[\text{grid}_x][\text{grid}_y] = $ vector of site-specific stationary values $s_i$
        Set $\text{A}[\text{grid}_x][\text{grid}_y]$ as vertex of $G$
    **end for**
    **for** $f_i \in F$ **do**
        **for** $n_i \in N$ **do**
            Let $\text{start}_x, \text{start}_y$ be the starting coordinates of a weighted directed edge in $G$
            $\text{start}_x, \text{start}_y = \text{grid}_x, \text{grid}_y$
            Recover $\text{end}_x, \text{end}_y$ from site-specific non-stationary value $n_i$.
            Create weighted directed edge in $G$ starting from vertex located at $(\text{start}_x, \text{start}_y)$
            and ending at vertex located at $(\text{end}_x, \text{end}_y)$ with weight of $|n_i|$.
        **end for**
    **end for**

**Output:** Geographically bound graph feature matrix grid $A$, Weighted Directed Graph $G$

---

For each of the data sources, we construct a set of 3D input "images" with dimensions of (rows, columns, filters). To construct a 5D tensor for the Keras ConvLSTM layer, we bundle all input frames over time into multiple samples. We bundle 24 consecutive frames into a single sample, where each frame represents information at a timestep with an hourly temporal frequency. Each bundle of 24 frames then represents a single day's worth of data.

The input data bundles are staggered such that the first sample consists of data from frames 1–24, the second sample consists of data from frames 2–25, and so on. In this way, we continue to preserve a continuous flow of temporal correlations among samples. By constructing this 5D tensor, we can transform the 26,304 3D input "images" (24 samples of 1096 days) into a 5D tensor of shape (26,304, 24, 40, 40, 10). The 10 filters in the 5D tensor consist of three filters for the MERRA-2 fire features (PBL height, surface temperature, and surface exchange coefficient for heat), 1 filter for the MODIS FRP imagery, one filter for the MAIAC MODIS AOD imagery, three filters for the TROPOMI data of air pollutants (nitrogen dioxide, carbon monoxide, and methane), one filter for the output of the GCN on meteorological data, and one filter for the ground-based PM2.5 sensor data from AQMIS2. Figure 11 provides a visualization of these input filters.

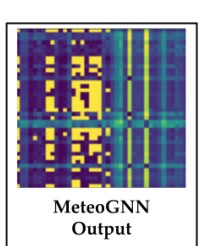 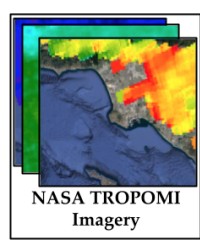 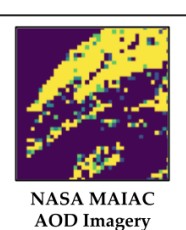 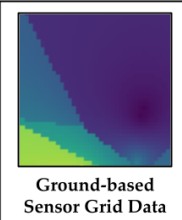 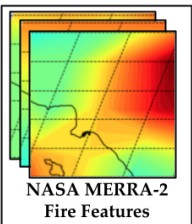 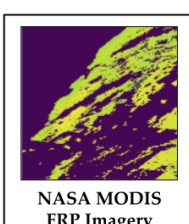

| MeteoGNN Output | NASA TROPOMI Imagery | NASA MAIAC AOD Imagery | Ground-based Sensor Grid Data | NASA MERRA-2 Fire Features | NASA MODIS FRP Imagery |

**ConvLSTM Input Filters**

**Figure 11.** Visualization of input filters for our ConvLSTM model.

To evaluate and test our model, we add a final Dense Keras layer with seven neurons to give a prediction for solely the seven PM2.5 sensor locations instead of a spatially continuous prediction of a 40-by-40-pixel grid over Los Angeles county [37]. We have the capability to produce spatially continuous predictions of PM2.5 with our current model,

but to evaluate against existing ground truth values with little to no measurement error or uncertainty, we restrict the prediction to monitoring sites available in the California ARB AQMIS2 portal.

## 3. Results

Our model predicts spatiotemporal PM2.5 in terms of micrograms per cubic meter ($\mu$g/m$^3$) at seven sensor locations in Los Angeles county hourly using 24 h of data in the past to predict 24 h of data in the future using meteorological data, wildfire data, remote-sensing satellite imagery, and ground-based sensor data. We use 1065 days of data from 1 January 2018 to 30 November 2020 as training data and evaluate our prediction on a test dataset of 744 samples (24 samples for 31 days) from 1 December 2020 to 31 December 2020. Figure 12 provides a visualization of the distribution and variance of the ground truth PM2.5 values for each sensor location.

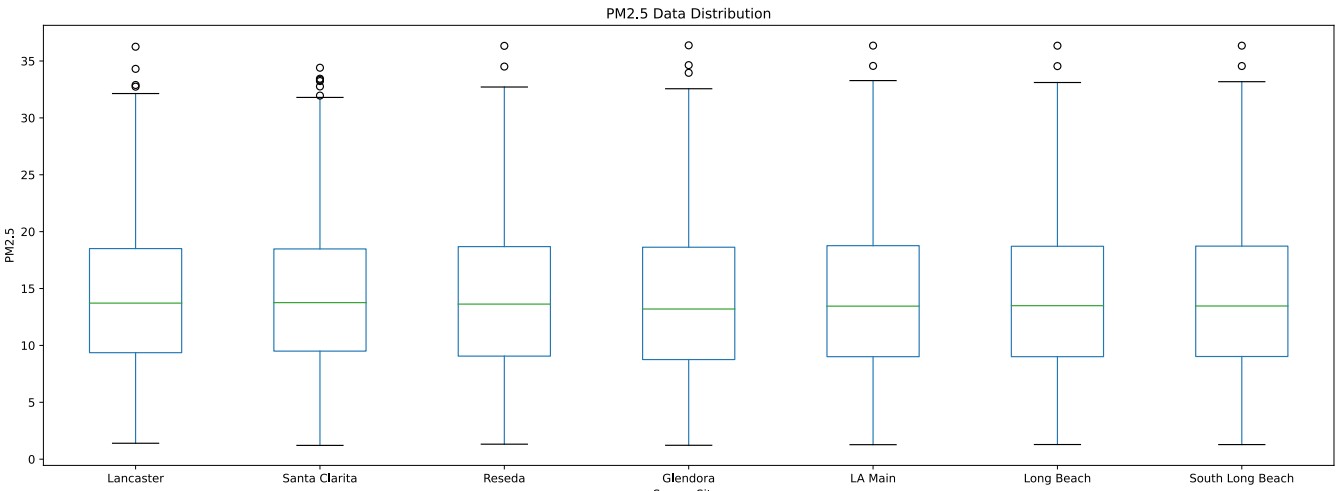

**Figure 12.** Data Distribution Plot of PM2.5 Ground Truth Sensors in LA County during Testing Timeframe (1 December 2020–31 December 31 2020).

To measure the accuracy of our model, we use the Root Mean Square Error (RMSE) and Normalized Root Mean Square Error (NRMSE) error. RMSE and NRMSE is calculated as

$$\text{RMSE} = \sqrt{\sum_{i=1}^{n} \frac{(\hat{y}_i - y_i)^2}{n}}; \ \text{NRMSE} = \frac{\text{RMSE}}{\bar{y}}$$

where $n$ is the number of observations, $\hat{y}$ is the predicted value, $y$ is the ground truth, and $\bar{y}$ is the mean of the test data.

Table 1 displays the prediction RMSE and NRMSE metric results for the first frame average and 24th frame average for each sensor location throughout the test set. Please note that first frame average error denotes the average error of the immediate next frame predicted using the previous 24 frames, while the 24th frame average error denotes the average error of the 24th of 24 frames using 48 frames earlier than the 24th frame. Since the first frame predictions uses more recent data to predict, the average first frame error is thus lower than the average 24th frame error.

Table 2 displays the prediction RMSE and NRMSE metric results on the first 24 frames of the testing set (24 h of 1 December 2020 prediction).

**Table 1.** RMSE (in µg/m³) and NRMSE error values averaged over 24 frame bundles (First Frame Averages and 24th Frame Averages) of test set for each sensor location.

| Metric | Sensor Location | Average Value | |
|---|---|---|---|
| | | 1st Frame | 24th Frame |
| RMSE | Lancaster | 0.753323 | 1.039245 |
| | Glendora | 0.747422 | 1.138471 |
| | Santa Clarita | 0.748873 | 0.937149 |
| | Reseda | 0.738090 | 0.875254 |
| | LA—Main St | 0.648110 | 0.773491 |
| | Long Beach | 0.631741 | 0.759828 |
| | Long Beach—RT 710 | 0.619384 | 0.738418 |
| NRMSE | Lancaster | 0.054438 | 0.2866 |
| | Glendora | 0.054642 | 0.063814 |
| | Santa Clarita | 0.054677 | 0.066814 |
| | Reseda | 0.053913 | 0.061014 |
| | LA—Main St | 0.048314 | 0.059641 |
| | Long Beach | 0.044283 | 0.053194 |
| | Long Beach—RT 710 | 0.040283 | 0.051743 |

**Table 2.** RMSE (in µg/m³) and NRMSE Error Values for first five frames of test set or roughly 10 days of data (5 December 2019–11 December 2019).

| Frame | Metric | |
|---|---|---|
| | RMSE | NRMSE |
| 1 | 0.671925 | 0.049715 |
| 2 | 0.678362 | 0.049751 |
| 3 | 0.673139 | 0.048904 |
| 4 | 0.689085 | 0.051835 |
| 5 | 0.713598 | 0.054311 |
| 6 | 0.677656 | 0.050077 |
| 7 | 0.719502 | 0.053545 |
| 8 | 0.771981 | 0.058694 |
| 9 | 0.904364 | 0.061504 |
| 10 | 0.673252 | 0.048378 |
| 11 | 0.907358 | 0.064156 |
| 12 | 0.672855 | 0.047664 |
| 13 | 0.847963 | 0.059228 |
| 14 | 1.073242 | 0.069844 |
| 15 | 0.709482 | 0.052143 |
| 16 | 0.756776 | 0.051116 |
| 17 | 0.748892 | 0.056426 |
| 18 | 0.672018 | 0.051460 |
| 19 | 0.674777 | 0.048313 |
| 20 | 0.678842 | 0.052491 |
| 21 | 0.684812 | 0.050102 |
| 22 | 0.866935 | 0.068929 |
| 23 | 0.765170 | 0.052922 |
| 24 | 1.042741 | 0.081660 |

Our results show significant improvement over current state-of-the-art deep-learning models on predicting spatiotemporal PM2.5 air pollution. Our testing set first frame prediction's percentage accuracy is 95.03%, which is a 60.3% decrease in hourly error from the leading implementations of the ConvLSTM model for PM2.5 prediction [16]. Our testing set's best frame accuracy over the first 24 h in the testing set is 95.10%. Moreover, our results show a 91% decrease in first frame error compared to our previous model using solely the ConvLSTM model on Sentinel-2 satellite imagery [38–44]. Our results also show a 43% decrease in first frame error compared to our previous model with a similar

architecture without wildfire data from NASA MODIS and MERRA-2. The averaged RMSE and NRMSE decrease over time with later frames, but this is expected as the nature of PM2.5 results in concentrations 24 h in the future being more correlated with 24 h in the past as compared to concentrations 48 h in the future.

## 4. Conclusions

In this paper, we use complex deep-learning models to accurately predict spatiotemporal PM2.5 in Los Angeles county over time in hourly temporal frequencies using meteorological data, wildfire data, remote-sensing satellite imagery, and ground-based sensor data. In designing our model, we include information on spatial and temporal correlations as well as meteorological features, wildfire patterns, smoke plumes, and related air pollutant matter data to understand, learn, and predict spatiotemporal PM2.5 air pollution.

We approach the complex task of predicting spatiotemporal PM2.5 through a deep-learning perspective. In our approach, we focus on developing robust deep-learning algorithms capable of decoding multisource big data in various formats. Thus, there are limitations for deep-learning models of this nature. We do not consider atmospheric physics or chemical mechanisms. There are certain local sources we do not consider, including the terrain and elevation of the study area or the traffic trends within Los Angeles. Although these sources may be indirectly present in the input meteorological data, we do not directly use such data. Furthermore, these local sources are often overshadowed in its contribution to PM2.5 when significant wildfire or smoke events occur in the vicinity. Thus, using both remote-sensing data and satellite imagery of wildfires, our approach is effective in understanding the effect of wildfire or smoke events to PM2.5 in a close range. For larger regional impacts of wildfires, we do not consider their chemical mechanisms or account for dilution.

We use various cutting-edge deep predictive models including the Graph Convolutional Network (GCN) and the Convolutional Long Short-Term Memory (ConvLSTM). We create a time-parameterized set of multidimensional weighted directed graphs to represent 17 meteorological features in 24 sensor locations within the greater Los Angeles county area through a novel algorithm. We then use the GCN architecture to perform convolution on neighborhoods of nodes and edges to interpolate dense meteorological graphs using spatiotemporal kriging. We also use unsupervised graph representation learning algorithms to create high-level embedding "images" of the dense meteorological graphs and use these high-level embeddings as input to the ConvLSTM model. In addition to the outputs from the GCN, we also supply validated ground-based PM2.5 sensor data in grid format, NASA MODIS MAIAC AOD remote-sensing satellite imagery, TROPOMI carbon monoxide, nitrogen dioxide, and methane remote-sensing data, MERRA-2 PBL height, surface air temperature, and surface exchange coefficient for heat fire features, and MODIS FRP remote-sensing satellite imagery as input to the ConvLSTM. We calculate the RMSE and NRMSE error values of the predicted PM2.5 values over the first 24 frames as well as the averaged RMSE and NRMSE error values of the predicted sample for each sensor location. We find that our results show significant improvement upon current research in the field using spatiotemporal deep predictive algorithms.

## 5. Future Work

In the future, we hope to predict spatially continuously across Los Angeles. We can achieve better results on grid-based predictions by implementing advanced interpolation models on the ground-based PM2.5 sensor input data [45]. We also hope to calculate and account for the data fusion under uncertainty error for ground-based sensor measurements to ensure the validity of recorded values. Doing this will allow us include low-cost individually maintained ground-level sensor data as inputs and predictive targets to increase the spatial resolution of predictions.

This research can also extend further than Los Angeles county and predict an array of pollutants including carbon monoxide, ozone, nitrogen dioxide, and sulfur dioxide. This

work can be used to inform and assist researchers in various disciplines on the movement of PM2.5 along temporal and spatial coordinates.

**Author Contributions:** Conceptualization, P.M., K.N., D.C., C.F.C., N.A., J.H., and M.P.; methodology, P.M., K.N., and M.P.; software, P.M., K.N., N.A., and M.P.; validation, P.M., K.N., D.C., C.F.C., N.A., J.H., and M.P.; formal analysis, P.M., K.N., and M.P.; investigation, P.M., K.N., and M.P.; resources, D.C., C.F.C., N.A., J.H., and M.P.; data curation, P.M., K.N., D.C., C.F.C., J.H., and M.P.; writing—original draft preparation, P.M.; writing—review and editing, P.M., K.N., and M.P.; visualization, P.M.; supervision, P.M., K.N., J.H., and M.P.; project administration, D.C., C.F.C., J.H., and M.P.; funding acquisition, J.H. and M.P. All authors have read and agreed to the published version of the manuscript.

**Funding:** This research is supported by NASA and the City of Los Angeles through the Predicting What We Breathe project.

**Data Availability Statement:** Data and methods used in the research have been presented in sufficient detail in the paper.

**Conflicts of Interest:** The authors have no conflict of interest to declare.

## Appendix A

**Table A1.** Data summary table of input data source databases, instruments, and data source types.

| Data | Database | Instrument | Data Source Type |
|---|---|---|---|
| PM2.5 | CARB AQMIS2 | Monitoring Station | Ground Sensor Data |
| Meteorological Features | Iowa Environmental Mesonet | Monitoring Station | Ground Sensor Data |
| MAIAC AOD | NASA AppEARS | NASA MAIAC (MCD19A2) | Satellite Imagery |
| MODIS FRP | NASA AppEARS | NASA MODIS (MOD11A1) | Satellite Imagery |
| PBL Height | NASA EarthData | NASA MERRA-2 (M2T1NXFLX) | Satellite Imagery |
| Surface Air Temperature | NASA EarthData | NASA MERRA-2 (M2T1NXFLX) | Satellite Imagery |
| Surface Exchange Coefficient for Heat | NASA EarthData | NASA MERRA-2 (M2T1NXFLX) | Satellite Imagery |
| Carbon Monoxide (CO) | NASA EarthData | ESA Sentinel-5P TROPOMI | Remote-sensing Data |
| Methane ($CH_4$) | NASA EarthData | ESA Sentinel-5P TROPOMI | Remote-sensing Data |
| Nitrogen Dioxide ($NO_2$) | NASA EarthData | ESA Sentinel-5P TROPOMI | Remote-sensing Data |

**Table A2.** METAR Meteorological Features for each of the 24 ASOS sites within Los Angeles county collected from Mesonet.

| Meteorological Feature | Unit | Stationary/Non-Stationary |
|---|---|---|
| Air Temperature | F | Stationary |
| Dew Point | F | Stationary |
| Relative Humidity | % | Stationary |
| Heat Index/Wind Chill | F | Stationary |
| Wind Direction | ° | Non-Stationary |
| Wind Speed | mph | Non-stationary |
| Altimeter | in | Stationary |
| Sea Level Pressure | mb | Stationary |
| 1 Hour Precipitation | in | Stationary |
| Visibility | mi | Stationary |
| Wind Gust | mph | Stationary |
| AQI | N/A | Stationary |
| Peak Wind Gust | mph | Non-Stationary |
| Peak Wind Direction | ° | Non-Stationary |
| Cloud Height Level 1 | ft | Stationary |
| Cloud Height Level 2 | ft | Stationary |
| Cloud Height Level 3 | ft | Stationary |

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
