# Peer review of "PM2.5 Air Pollution Prediction through Deep Learning Using Multisource Meteorological, Wildfire, and Heat Data"

_atmosphere, doi:10.3390/atmos13050822_

Round 1

Reviewer 1 Report

PM2.5 Air Pollution Prediction through Deep Learning using Multisource Meteorological, Wildfire, and Heat Data is an important area of research. Keeping in view the state of manuscript I recommend it can be published in the “atmosphere” journal.  My comments are as follows:

(1)The introduction of the manuscript should focus on a brief description of the existing study research of PM2.5 prediction, not air pollution;

(2)In the introduction of the manuscript, the deep learning algorithm used in this paper should be put into the section 2 Methodology;

(3)The data (such as FRP, AOD, meteorology, etc.) scales used in the manuscript vary greatly. The author should clarify the scale of various data and how to achieve the consistency of each data scale.

(4)The meteorological data predicted by GCN algorithm in the manuscript also need to be evaluated for accuracy;

(5)The manuscript should analyze and compare the depth algorithm in this paper to predict PM2 5 and existing literature depth algorithm to predict PM2 5 accuracy.

Author Response

Dear Reviewer, 

We sincerely thank you for your valuable time and comments on our submitted manuscript. Please find our attached revised manuscript entitled "PM2.5 Air Pollution Prediction through Deep Learning using Multisource Meteorological, Wildfire, and Heat Data" by Pratyush Muthukumar, Kabir Nagrecha, Dawn Comer, Chisato Fukuda Calvert, Navid Amini, Jeanne Holm, and Mohammad Pourhomayoun. This is the revision of our earlier submission with the same
title. Please find our point-by-point responses to your comments below. We thank you again for your valuable time and expertise in writing these insightful comments which have been instrumental in elevating the quality of our revised manuscript. 

(1) We appreciate this suggestion and we have made the appropriate changes to our revised manuscript. In the Introduction section, we included a paragraph on the current state-of-the-art deep learning research on PM2.5 prediction (See Page 2 Lines 40-51).

(2) We agree with this suggestion and have restructured our revised manuscript such that the description of our two stage GCN-ConvLSTM model (including the equations) have been moved from Section 1 (Introduction) to Section 2.1 (Model Architecture). 

(3) We understand the concerns raised and have clarified this in our revised manuscript. It is true that the raw meteorological data we use contains data of various scales and units (e.g Temperature in Fahrenheit, Relative Humidity in %, Wind Speed in MPH, Precipitation in Inches, etc), but we solve this issue by instead converting each individual hourly measurement into a percentile value of the daily maximum. In this way, we do not utilize the raw scales and units of the individual measures. Further explanation of this process can be found in Page 8 Lines 245-252. 

(4) The GCN algorithm used to perform interpolation on the weighted directed graphs containing meteorological data follows directly from the algorithms proposed by Wu et al in "Inductive Graph Neural Networks for Spatiotemporal Kriging". Reference [25] contains the link to the paper where in-depth error analysis is conducted. For the sake of brevity and due to the fact that after this GCN algorithm has interpolated the grids, we feed it to an unsupervised graph representation learning algorithm, we did not include further error analysis of the meteorological GCN step in our paper. 

(5) We agree with this suggestion and, in our revised manuscript, have included the mean accuracy of predictions from the state-of-the-art deep learning approach which can be compared against our results described in Table 1 and 2. Our results show improvement in accuracy, spatial resolution, and temporal resolution over current best-performing deep learning studies in predicting PM2.5. 

Best Regards,
Pratyush Muthukumar  

Reviewer 2 Report

This study presents the development and results of spatiotemporal PM2.5 predictions using a Graph Convolutional Network (GCN) architecture coupled to a Convolutional Long Short-Term Memory (ConvLSTM) architecture. The novelty in the approach is in the use of sequentially learned meteorological feature correlations obtained using the GCN which are used as input for the ConvLSTM in addition to ground-based air pollutant sensor data, remote-sensing satellite imagery and wildfire data to predict spatiotemporal PM2.5 air pollution.

The results indicate that the RMSE and NRMSE error values for the predicted PM2.5 values show up to 60.3% decrease in the hourly error values compared to previous studies. The paper is well written, the arguments, methodology and results presented properly.

The study will benefit the community and contribute to knowledge in this study area and will have a lasting impact.

A number of minor amendments need to be addressed:

Page 2:

This statement is incorrect:

“Within Los Angeles, there are over 27 million tons of atmospheric nitrogen dioxide, which doubles the amount of the next leading U.S. city (LA Times, 2019). It is evident that finding an effective and reliable solution to reducing ambient air pollution will drastically improve global health and well-being.”

Referring to the particular Los Angeles Times article [see https://www.latimes.com/environment/story/2019-11-15/nox-pollution-los-angeles-air-quality], this is a more accurate picture from the article “However, since 2010, the decline slowed and nearly flattened, dropping only to 25 metric tons in 2013. That left Los Angeles with NOx levels high above second-place Chicago (with nearly 17 metric tons) and third-place Detroit (just under 12 metric tons).” Thus Los Angeles has 1.5 times the atmospheric nitrogen dioxide level of Chicago, not double.

Referencing:

Please check the referencing style recommended for the journal but the references to the online articles are incomplete and do not indicate on which day[s] they were accessed. These include:

Erik Boye Abrahamsen, Ole Magnus Brastein, and Bernt Lie. Machine Learning in Python for Weather Forecast based on Freely Available Weather Data. 2018.

CARB. Air quality and meteorological information system, 2021. URL https: //www.arb.ca.gov/aqmis2/aqmis2.php.

Keras. Tf.keras.layers.convlstm2d, 2021a. URL https://www.tensorflow.org/ api_docs/python/tf/keras/layers/ConvLSTM2D.

Keras. Keras documentation: Dense layer, 2021b. URL https://keras.io/api/ layers/core_layers/dense/

LA Times. New satellite measurements show how polluted los angeles’ air really is, Nov 2019. URL https://www.latimes.com/environment/story/ 2019-11-15/nox-pollution-los-angeles-air-quality.

McGill University. Air pollution: The silent killer called pm2.5, Apr 2021.

NASA. Mod11a1 v006, 2021. URL https://lpdaac.usgs.gov/products/ mod11a1v006/.

National Geographic. Air pollution, Oct 2012.

NCAR. Nasa’s merra-2 reanalysis, 2019. URL https://climatedataguide. ucar.edu/climate-data/nasas-merra2-reanalysis.

  1. With a premature death every five seconds, air pollution is violation of human rights, 2019. United Nations.

Grammar/Spell recommendations:

Page 10

You could change “To use these meteorological features within model, we must transform the array” to “To use these meteorological features within the model, we must transform the array”

Page 14

You could change “PBL Height is also infulenced by” to “PBL Height is also influenced by”

Page 16

You could change “Finally, we utilize the MERRA-2 surface exchange coefficient for heat feature.” to “Finally, we utilize the MERRA-2 surface exchange coefficient for the heat feature.”

Author Response

Dear Reviewer, 

We sincerely thank you for your valuable time and comments on our submitted manuscript. Please find our attached revised manuscript entitled "PM2.5 Air Pollution Prediction through Deep Learning using Multisource Meteorological, Wildfire, and Heat Data" by Pratyush Muthukumar, Kabir Nagrecha, Dawn Comer, Chisato Fukuda Calvert, Navid Amini, Jeanne Holm, and Mohammad Pourhomayoun. This is the revision of our earlier submission with the same title. Please find our point-by-point responses to your comments below. We thank you again for your valuable time and expertise in writing these insightful comments which have been instrumental in elevating the quality of our revised manuscript. 

- In regards to the statement on page 2 regarding the LA Times article, we thank the reviewer for pointing this out and have made changes accordingly in our revised manuscript (See Page 1 Line 27). 
- In regards to the comments on referencing, we agree with the reviewer and have updated the references noted to include the URL and the date accessed in accordance with the MDPI journal reference format guidelines.
- In regards to the grammatical recommendation on page 10, we have updated the revised manuscript with the noted suggestion (See Line 245). 
- In regards to the grammatical recommendation on page 14, we have updated the revised manuscript with the noted suggestion (See Line 349). 
- In regards to the grammatical recommendation on page 16, we have updated the revised manuscript with the noted suggestion (See Line 361). 

Best Regards,
Pratyush Muthukumar

Reviewer 3 Report

General Comments: (overall quality)

The paper is written in an easy to read. The figures and tables are in general good. In my opinion, the introduction and methodology sections should be improved, according to the specific comments listed below, all of which are actually only minor revisions. These aspects would make your manuscript more convincing.

Specific Comments:

Section 1 (Introduction)

1. This section is lengthy and very detailed. Some parts, such as the description of the deep learning architectures could be moved into an Appendix Section.

2. Do you know previous studies that promoted integration of data from in situ and remote sensing measurements using neural network architecture? I recommend that you provide an overview about previous approaches to the use of neural networks in air quality studies and predictions.

Section 2 (Methodology)

1. The authors must give more details about the ground based air quality monitoring in Los Angeles county and surroundings. What are the typical aspects of atmospheric circulation for Los Angeles county ?

2. What are the advantages/benefits of the deep learning technique proposed in this work compared to neural networks technique used in previous studies? Please comment on that.

Author Response

Dear Reviewer, 

We sincerely thank you for your valuable time and comments on our submitted manuscript. Please find our attached revised manuscript entitled "PM2.5 Air Pollution Prediction through Deep Learning using Multisource Meteorological, Wildfire, and Heat Data" by Pratyush Muthukumar, Kabir Nagrecha, Dawn Comer, Chisato Fukuda Calvert, Navid Amini, Jeanne Holm, and Mohammad Pourhomayoun. This is the revision of our earlier submission with the same title. Please find our point-by-point responses to your comments below. We thank you again for your valuable time and expertise in writing these insightful comments which have been instrumental in elevating the quality of our revised manuscript. 

Section 1 Comments: 

(1) We agree with this suggestion and have made the necessary changes in our revised manuscript. We have moved the definitions of our two stage GCN-ConvLSTM model (and the equations) from Section 1 (Introduction) to Section 2.2 (Model Architecture), which significantly reduces the length of the Introduction section. 

(2) We appreciate this suggestion and have added a paragraph within the Introduction section of our revised manuscript on the current state-of-the-art deep learning study for predicting PM2.5 (See Page 2 Lines 40-51). 

Section 2 Comments:

(1) We thank the reviewer for pointing this out. In our revised manuscript, we have added a paragraph within Section 2 (Methodology) describing the atmospheric circulation environment in and around the LA Basin (See Page 3 Lines 94-101). 

(2) We appreciate this suggestion and have included a paragraph within the Introduction section describing how the main advantages of our proposed model against the state-of-the-art deep learning model for predicting PM2.5 includes predictions with higher spatial resolution, temporal resolution, and accuracies (See Page 2 Lines 40-51). 

Best Regards,
Pratyush Muthukumar  

Reviewer 4 Report

I begin with an example. TROPOMI does not take pictures as authors mention in abstract. Reading the rest manuscript I make the opinion that authors are not familiar with the right use of many terms even the “air pollution”. For example, they mention that without know how, many air pollutants, NO2, CO, O3, CH4, is highly correlated with PM2.5. They do not justify their claim or describe how this works in the area they study. These facts give me the impression that authors are not familiar with the study of atmosphere, they are only familiar with neural networks.

The development of the prediction model is totally unexplainable. Authors are based only on mixing some kind of data and producing some results about PM2.5 levels. Their model is totally a black box. No atmospheric physics and chemistry mechanisms are considered.

I strongly believe that studying atmosphere through artificial intelligence ‘s methods and techniques without understanding what is going on in the real world  is totally wrong so I suggest the rejection of the manuscript.

Author Response

Dear Reviewer, 

We sincerely thank you for your valuable time and comments on our submitted manuscript. Please find our attached revised manuscript entitled "PM2.5 Air Pollution Prediction through Deep Learning using Multisource Meteorological, Wildfire, and Heat Data" by Pratyush Muthukumar, Kabir Nagrecha, Dawn Comer, Chisato Fukuda Calvert, Navid Amini, Jeanne Holm, and Mohammad Pourhomayoun. This is the revision of our earlier submission with the same title. Please find our point-by-point responses to your comments below. We thank you again for your valuable time and expertise in writing these insightful comments which have been instrumental in elevating the quality of our revised manuscript. 

- In regards to our claim that other air pollutants are highly correlated to PM2.5, we understand the reviewer's concern and would like to note that we provide evidence from a study showing that a large portion of ambient PM2.5 is generated through the chemical reactions of atmospheric nitrogen dioxide (See Page 5, Lines 164-165)
- In regards to our familiarity with the study of the atmosphere, we would like to note that we approach the task of predicting spatiotemporal PM2.5 through the lens of artificial intelligence and deep learning. It is true that the authors of this paper are primarily computer scientists, and as such are not as well versed in the atmospheric sciences. However, as recent deep learning studies on predicting atmospheric and ground-level pollutants show, the capabilities of modern deep learning architectures are able to perform to the level of and exceed performances of chemical and physical models which take into account the nuances of atmospheric science in solving similar tasks. As such, although there may be certain well-known downsides to applying deep learning, such as a lack of interpretability, the positive performance results of these "black box" models certainly account for these drawbacks noted. However, to address the comments made, we included a paragraph to Section 2 (Methodology) describing the atmospheric circulation specifics in and around the LA Basin (See Page 3, Lines 94-101). 
- In regards to our model being a black box, this is a comment which could be applied to all modern deep learning approaches to predicting atmospheric or ground level pollution. Although it is well known that deep learning approaches are often, as noted "black boxes", and thus suffer from a lack of interpretability, there is no denying the predictive power of modern deep learning, especially in the application of predicting pollutants such as PM2.5. The current state-of-the-art results for predicting PM2.5 are currently black box deep learning models. Although these approaches do not take into account the chemical and physical mechanisms at play, they are still able to outperform traditional models considering these factors. While we agree that including such information can only serve to help our model with predicting PM2.5, we approached the task of PM2.5 prediction in the regime of modern deep learning, and as such, when developing our model, we were more invested in collecting multisource data with a variety of features.

Best Regards,
Pratyush Muthukumar

Round 2

Reviewer 4 Report

I am glad that authors agree with my comment that they are unfamiliar with the field they work in this manuscript. I leave the evaluation of this point to the Editor.

The revised manuscript has no changes concerning my major comments about the unexplainable nature of the developed model. They have not even changed the wrong terminology. For example, maps presenting plotted data are not considered as satellite images. Satellite images are something else. To my knowledge, this terminology has never been used and it is very confusing for the readers, especially the ones to which this manuscript is addressed, the atmospheric scientists.

The evidence they claim they refer is not adequate. It says nothing that “a large portion of ambient PM 2.5 is generated through the chemical reactions of atmospheric nitrogen dioxide.” In addition, the reference they give is almost 15 years old. Great progress has been made since then in atmospheric chemistry and physics.

The target of atmospheric scientists is to understand, not just to predict something. In their efforts, they produce a “first generation” model based on the knowledge they have that time, test it, study its weaknesses, gain knowledge from deeper studying, improve it, test it again and go on. But this is the way we have been here today.

I agree with authors response about the nature of deep learning that is why I believe deep learning should be used with high caution in science. Finally, this is a scientific journal and the manuscript add nothing in science.

For these reasons, I propose the rejection of the manuscript.

Minor comments

Abbreviations are defined the first time they are mentioned in the manuscript.

Author Response

To Reviewer,

We thank you for your valuable time and comments on our submitted manuscript. Please find our attached revised manuscript entitled "PM2.5 Air Pollution Prediction through Deep Learning using Multisource Meteorological, Wildfire, and Heat Data" by Pratyush Muthukumar, Kabir Nagrecha, Dawn Comer, Chisato Fukuda Calvert, Navid Amini, Jeanne Holm, and Mohammad Pourhomayoun. This is the revision of our earlier submission with the same title. Please find our point-by-point responses to your comments below. We thank you again for your valuable time and expertise in writing these insightful comments which have been instrumental in elevating the quality of our revised manuscript.

In regards to the comment about terminology regarding satellite imagery, we have revised our manuscript and removed all occurrences of the term "satellite imagery" used with the TROPOMI data source, and replace it with "remote-sensing data". We also change the figure caption of Figure 6, which now refers to the plotted TROPOMI data as remote-sensing data instead of satellite imagery.

In regards to the comment about the reference to the contribution of nitrogen dioxide on PM2.5, we find that multiple sources, past and present, note that NO2 along with other NOx  reacts with other chemicals in the air to form both particulate matter and ozone.

Sources:

  • Wang, Yuzhou, et al. "Spatial decomposition analysis of NO2 and PM2. 5 air pollution in the United States." Atmospheric Environment 241 (2020).
  • Hodan, William M., and William R. Barnard. "Evaluating the contribution of PM2. 5 precursor gases and re-entrained road emissions to mobile source PM2. 5 particulate matter emissions." MACTEC Federal Programs, Research Triangle Park, NC (2004).

However, since we do acknowledge that we are not experts in atmospheric physics or chemistry, we have removed the statement “a large portion of ambient PM 2.5 is generated through the chemical reactions of atmospheric nitrogen dioxide” from our revised manuscript. Instead, we simply note that other air pollutants can be correlated to PM2.5. (See Page 5 Line 162-163)

We have also added a sentence in our abstract and a paragraph in the Conclusion (Section 4) noting the limitations of our approach. We acknowledge that we do not consider the atmospheric physics or chemical mechanisms in our deep learning model. (See Lines 6-7 and Lines 462-474)

In regards to the comment about our manuscript's contribution to science and scientific journals, we would like to note that we are submitting this manuscript to the Atmosphere Special Issue in "Big Data and Artificial Intelligence for Air Quality Assessment and Forecasting". Within the Special Issue Information page, the Editors note that "this Special Issue (SI) aims to discuss the role and applicability of top BD [Big Data] technologies in the evaluation and prediction of AQ based on massive air pollution measurement data provided by sensor networks. Scientists and researchers are invited to contribute to this SI by submitting manuscripts (research papers, communications, review articles) describing the fundamentals, underlying models and algorithms, and practical cases of analytical BD and AI technologies for the assessment and forecasting of AQ in real scenarios." Clearly, our manuscript contributes to the goals of this journal by proposing the development and application of novel deep learning architectures (e.g. GCN, ConvLSTM) indicated through our model's significant improvement over existing research in predicting PM2.5 using spatiotemporal deep predictive algorithms.

Best regards,
Pratyush Muthukumar
